# Formalizing the Binding Problem

**Lianghuan Huang** [*1]   **Yihao Li** [*2]   **Saeed Salehi** [3]   **Yingshan Chang** [4]   **Ansh Soni** [5]   **Konrad P. Kording** [6]

## Abstract

Representations of the world, arguably, contain information about features (e.g. something is blue, something is a circle) but also information about which features are part of the same object (e.g. the circle is blue), which we call binding information. Any system with the ability to understand scenes with multiple objects must be able to solve the binding problem: it needs to know which features belong together. However, despite work showing that Vision Transformers (ViTs) know which patches belong together, it is not known whether current deep learning models learn to exhibit binding information, i.e., for features. We may believe that there is not much binding information, after all misattributing features to wrong objects is a common failure of ViT-based architectures, especially in scenes with objects sharing features. Here we formalize the binding problem with an information-theoretic approach, and introduce a probing method to measure binding information in model representations. We perform experiments on ViTs, measuring binding from different components of the architecture, such as the image summary token `[CLS]` or the spatial tokens. We use datasets with different binding challenges, such as feature sharing, occlusion, and natural features, while comparing the performance of several pre-trained ViTs. Overall, our research demonstrates binding as a key ingredient to strong visual recognition and reasoning.[1]

---

[*]Equal contribution   [1]Department of Physics and Astronomy, University of Pennsylvania, Philadelphia, PA, USA [2]Department of Computer and Information Science, University of Pennsylvania [3]Machine Learning Group, Technical University of Berlin, Berlin, Germany [4]Language Technology Institute, Carnegie Mellon University, Pittsburgh, PA, USA [5]Department of Psychology, University of Pennsylvania [6]Department of Neuroscience, University of Pennsylvania, Philadelphia, PA, USA. Correspondence to: Konrad P. Kording <kording@seas.upenn.edu>.

*Proceedings of the 43rd International Conference on Machine Learning*, Seoul, South Korea. PMLR 306, 2026. Copyright 2026 by the author(s).

[1]Code available at: https://github.com/KordingLab/formalizing-the-binding-problem.

## 1. Introduction

Binding is the ability to tell which features belong to which objects in a multi-object scene. Humans naturally bind features to objects. This ability is so effortless that we rarely recognize binding as a computational problem in its own right. In the visual cortex, distinct neuronal populations are tuned to specific features (e.g., color, orientation, motion), but in multi-object scenes such feature-based encoding alone does not specify which features belong together as parts of the same object. Nevertheless, the brain achieves binding with extraordinary precision, and a range of mechanisms have been proposed to account for this capacity (Treisman & Gelade, 1980; von der Malsburg, 1981; Zhang et al., 2020) (more in Section A), but it remains largely an open problem (Yu & Lau, 2023).

Binding is just as equally important for artificial models. Compositional learning, a hallmark of generalization and reasoning for deep learning models (Uselis et al., 2025; Lake & Baroni, 2018; Hupkes et al., 2020; Wiedemer et al., 2023), relies crucially on the ability to bind the compositionally learned concepts into novel combinations (e.g., having learned the concepts of animals, body parts, and motion, can a model understand a flying penguin?) (Greff et al., 2020). However, artificial vision and vision-language models perform markedly worse at binding than the brain. In multi-object scenes, especially when objects are cluttered and/or share features, the ability of modern vision and vision–language models to correctly identify objects and their corresponding features drops noticeably (Campbell et al., 2025; Zhang et al., 2024; Lewis et al., 2024; Yuksekgonul et al., 2023; Assouel et al., 2025b). For instance, (Campbell et al., 2025) shows that the ability of a vision-language model to accurately describe objects and their features monotonically decreases as the number of feature-sharing object triplets increases in a scene (Figure 1), with similar trends observed in object counting, search, and analogy reasoning tasks. As another example, (Zhang et al., 2024) prompts vision-language models to describe objects in a given block of a Raven matrix (Figure 1). The model instead combines elements from adjacent blocks, attributing them to the same object. They further show that segmenting the grid into separate images before passing them in significantly reduces such errors. The failure of artificial vision models in binding as compared to the brain raises a

natural question: to what extent do artificial models reliably learn binding information? Is there a framework that can precisely define binding information learned by a model, and is there a formal way of measuring binding as defined by the theory?

Using an information-theoretic approach, we define binding as the information a representation contains about which objects are present or absent among all possible objects. We then introduce several variants of this definition that separate binding information from feature information, or normalize over dataset binding uncertainty priors. We then introduce a probing method to measure binding information defined as such. Finally, we apply this framework to state-of-the-art Vision Transformers, measuring the binding information learned in the image-level summary token [CLS] and the full set of spatial tokens, on datasets with different binding challenges. Overall, our contributions are as follows:

- We introduce an information-theoretic framework that defines binding and a probing method for measuring binding information in model representations.

- We show that the image-level summary token [CLS] encodes less than half of all binding information in a multi-object scene; for the binding information it does encode, it is largely structured quadratically.

- We show that binding is encoded almost perfectly in the full set of spatial tokens, decodable using a simplified attention probe.

- We perform probing experiments on a variety of models and datasets with different binding challenges, and analyze the binding information learned under each scenario.

## 2. Methods

Here we introduce an information-theoretic framework of formalizing binding, and show how binding information can be measured by probes. A scene has a set of features that are present. For the scene to make sense there are also objects, which are characterized by their specific conjunctions of features. And then there is the overall scene which is a collection of potential objects that are present or absent. Binding information is the information about which objects are present and absent.

### 2.1. Defining Binding Information

**Definition 2.1** (Features). Let the $f_i$ be discrete features. For example, they could be red, square, smooth, human, running, velocity=3m/s, occluded, top-left, etc.

Let $\mathcal{F} = \{f_1, f_2, \cdots, f_n\}$ be a finite set of all features of interest.

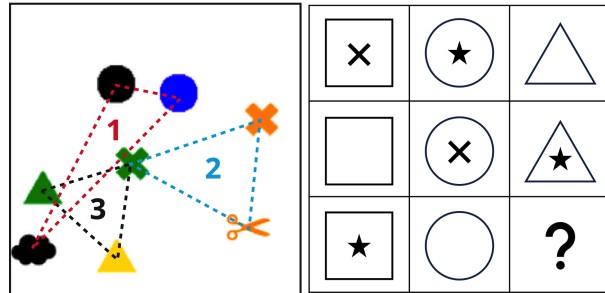

*Figure 1.* **Binding failures. Left:** (Campbell et al., 2025) shows that the ability of a vision-language model to accurately describe objects and their features in a scene degrades monotonically as the number of feature-sharing triplets (or the total number of objects) increases in a scene. **Right:** (Zhang et al., 2024) prompts vision-language models to describe grid patterns in the Raven matrix. For the middle-right block, the model outputs "a triangle with an X inside," combining elements from the middle-center and the middle-right blocks. They further show that segmenting the grid into separate images before passing them in significantly reduces such errors. Figures reproduced from the papers.

**Definition 2.2** (Feature code). One way of looking at a scene is in terms of existence of features. Some of the features may be shared by multiple objects (e.g. a blue bag and a blue hat). But the feature existence vector $F$ is useful to characterize which features are present. We define the feature code of scene as a finite random vector

$$F = [F_1, F_2, \ldots, F_n],$$

where

$$F_i = \begin{cases} 1 & \text{if feature } f_i \text{ is present in the scene,} \\ 0 & \text{otherwise.} \end{cases}$$

**Definition 2.3** (Objects). While $F$ does transmit which features are present, it does not describe how features jointly make up objects. An object can be described by the collection of its features. We posit that each object corresponds to a *distinct* subset of $\mathcal{F}$.

$$o_i \rightarrowtail \{f_a, f_b, \ldots\} \subseteq \mathcal{F}$$

For example, $o_i$ can be "red, blue, smooth cube," "human running at velocity=3m/s," or "top-left occluded triangle."

*Remark* 2.4. We posit that no two objects can correspond to the exact same set of features. In practice, *location* is often a discriminator for objects that may share all other features. For this reason, we consider the object to feature subset mapping a surjective map. More formally, let $\mathcal{O} = \{o_1, o_2, \cdots, o_n\}$ be a finite set of all objects of interest. Then there exist a *surjective* mapping $\psi$ such that

$$\psi : \mathcal{O} \rightarrowtail \mathcal{P}(\mathcal{F}),$$

where $\mathcal{P}(\mathcal{F})$ denotes the power set of $\mathcal{F}$.

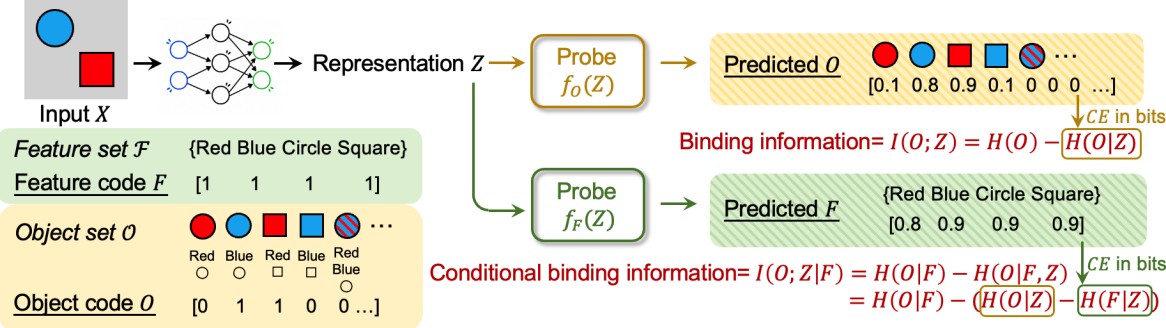

*Figure 2.* **Binding theory and probing framework.** We first define the space of features and objects of interest, as shown in features set $\mathcal{F}$ and object set $\mathcal{O}$ (Definitions 2.1, 2.3). Each object is a combination of features from the feature set (Definition 2.3, Remark 2.4). From there, we define the feature code $F$ and object code $O$, which are random vectors that denote the presence and absence of each feature and object in a scene $X$. Binding information $I(O; Z)$ is the information of the object code in the representation $Z$ (Definition 2.10). Conditional binding information is the information of the object code in the representation *beyond what can be explained by features*, hence $I(O; Z \mid F)$ (Definition 2.13). Using information theory, the $I$ terms can be decomposed into entropies $H$. $H(O \mid Z)$ and $H(F \mid Z)$ are the uncertainties of the object and feature codes in the representation, which can be estimated by training object code probe $f_O(Z)$ and feature code probe $f_F(Z)$ on the representation with ground truth labels of the object and feature codes (Section 2.2). The figure shows hypothetical probe-predicted logits for each element of the object and feature code. The cross-entropy losses of the object and feature probes are provable estimates of $H(O \mid Z)$ and $H(F \mid Z)$ (Theorem 2.20, Lemma 2.21). The remaining $H(O)$ or $H(O \mid F)$ terms are dataset priors which can be independently calculated from the distribution of the $F$ and $O$ in the dataset (Section 2.3). Combining these terms, we arrive at a probe-estimated value of binding information and conditional binding information.

**Definition 2.5** (Object code). An object code of a scene is a random vector

$$O = [O_1, O_2, \dots]$$

where

$$O_i = \begin{cases} 1 & \text{if object } o_i \text{ is present in the scene,} \\ 0 & \text{otherwise.} \end{cases}$$

*Remark* 2.6. To bind is to specify the object code of a scene, i.e., which objects exist and which objects do not (Definition 2.10). One potential difficulty of binding is that objects may share some (though not all) features with each other. For example, a scene may contain a red square, a blue circle, a red circle, but not a blue square. Their shared features can lead to binding errors under particular encoding schemes, especially when object encoding is correlated with feature encoding (Campbell et al., 2025; Greff et al., 2020; Treisman & Schmidt, 1982).

*Remark* 2.7. We note that our definition of object code is fully agnostic of the specific encoding scheme of objects of a model. While we *define* objects in feature-conjunctive ways (Definition 2.3), they may be encoded in many ways including tensor-product of features (Smolensky, 1990), slots (Locatello et al., 2020), object files (Kahneman et al., 1992), etc. For this reason, learning features well does *not* necessarily make learning objects easy for a model: in our definition, we do not *assume* their correlation or causation (although there usually *is* some correlation, see Definition 2.13).

*Remark* 2.8. The distinction between features and objects depends on our abstraction level. Objects can be features, and features can be objects, under different feature and object sets $\mathcal{F}, \mathcal{O}$. For example, "human" may be a feature among "chimpanzee" and "gorilla," but it may be an object when "mammal," "bipedal," "vertebrate" are features.[2]

*Remark* 2.9. Assuming access to the object-to-feature-subset map $\psi$ (Remark 2.4), for any particular scene, $F$ is a deterministic function of $O$. That is, once we know which objects exist in a scene, we also know which features exist, by virtue of the object-to-feature map.

We consider a model $\Phi : \mathcal{X} \to \mathcal{Z}$ which maps scenes to learned representations. Let $X \in \mathcal{X}$ be a random variable for the scene, and $Z \in \mathcal{Z}$ be a random variable for the representation. We consider $F$ and $O$ to be deterministic functions of $X$, given an oracle scene observer. The model $\Phi$, however, may *lose* information of $F$ and $O$ in its representation of the scene $X$, i.e., they are irrecoverable even with the optimal decoder. We therefore define binding information in a representation $Z$ as follows:

**Definition 2.10** (Binding information). Let $I(A; B)$ denote mutual information between random variables $A$ and $B$.

We define binding information in the representation $Z$ of a scene, relative to object set $\mathcal{O}$ (and feature set $\mathcal{F}$), as

$$\mathbf{B}_{\mathcal{O}}(Z) := I(O; Z),$$

---

[2]This can even be true within a neural network: objects learned by one layer may become features for the next layer. Neural networks may even be considered as "binding networks."

in unit of *bits*. In plain language, it is the information contained in the representation about the object code, i.e., about which objects are present and absent in a scene.

*Remark* 2.11. Let $H(A)$ denote the entropy of the random variable $A$, and $H(A \mid B)$ be its conditional entropy on $B$.

$$\mathbf{B}_\mathcal{O}(Z) := I(O; Z) = H(O) - H(O \mid Z) \qquad (1)$$

From the lens of entropy, binding information is also the reduction in the uncertainty of object code $O$ ($H(O)$) due to knowledge of the representation $Z$.

**Definition 2.12** (Feature information). Similarly, we can define feature information in the representation $Z$ as $I(F; Z)$, relative to feature set $\mathcal{F}$ (and object set $\mathcal{O}$).

Note that feature and object information are generally not independent within a representation. In particular, a representation that encodes features well *can* make objects easier to decode. Sometimes, however, we wish to measure the binding information contained in a representation beyond what can be explained by feature information alone. This may be especially desirable when comparing models with different feature learning capabilities,[3] or when comparing scene distributions with different feature learning complexities.[4] Under these scenarios, we may wish to measure binding information while controlling for decodable feature information. This can be done naturally by conditioning binding information on knowledge of the feature code.

**Definition 2.13** (Binding information, conditioned on feature code). We define binding information in a representation $Z$, conditioned on feature code $F$, as

$$\mathbf{B}^*_{\mathcal{O},\mathcal{F}}(Z) := I(O; Z \mid F)$$

**Theorem 2.14.**

$$\mathbf{B}^*_{\mathcal{O},\mathcal{F}}(Z) = I(O; Z) - I(F; Z).$$

*Proof in Appendix C.*

*Remark* 2.15. We note that conditioning on $F$ does not result in a *counterfactual* measure of binding information *as it would if a feature code were supplied to the model during*

---

[3] For example, there are architectures that are specifically designed for binding or binding-like operations, but they may be limited feature learners, possibly due to the poor scalability of their binding-related inductive biases. Examples of this may include tensor product architectures (Smolensky, 1990), capsule networks (Sabour et al., 2017), early slot attention models (Locatello et al., 2020; Seitzer et al., 2023), neural module networks (Andreas et al., 2017), etc.

[4] For example, natural datasets may contain high-level semantic features that are difficult to learn, but learning the object code may not be more difficult than, say, synthetic datasets with synthetic features.

*inference*. The representation $Z$ is fixed, so we are *adjusting* binding information *post-hoc* during measurement, for the amount of feature information, as is shown here.

**Theorem 2.16.** $\mathbf{B}^*_{\mathcal{O},\mathcal{F}}(Z)$ *can be similarly written in entropy form:*

$$\begin{aligned}\mathbf{B}^*_{\mathcal{O},\mathcal{F}}(Z) &= H(O \mid F) - H(O \mid Z, F) \\ &= H(O) - H(O \mid Z) - H(F) + H(F \mid Z).\end{aligned} \qquad (2)$$

*Proof in Appendix C.*

Some scene distributions may have higher prior uncertainties in $O$, as measured by $H(O)$. Binding information for a distribution is, at best, $H(O)$, since

$$\mathbf{B}_\mathcal{O}(Z) := I(O; Z) = H(O) - H(O \mid Z) \le H(O).$$

When comparing the difficulty of binding across different scene distributions, we may wish to remove the scale effect of this prior $H(O)$. This can be done by normalizing over $H(O)$.

**Definition 2.17** (Binding information, normalized by prior uncertainty of object code). We define binding information normalized by the uncertainty of $O$ in the scene distribution as

$$\beta_\mathcal{O}(Z) := \frac{I(O; Z)}{H(O)} = 1 - \frac{H(O \mid Z)}{H(O)},$$

without units.

Similarly, a normalized measure for the conditional binding information is

$$\beta^*_{\mathcal{O},\mathcal{F}}(Z) := \frac{I(O; Z \mid F)}{H(O \mid F)} = 1 - \frac{H(O \mid Z, F)}{H(O \mid F)}.$$

*Remark* 2.18. The normalized binding information can therefore be interpreted as the *proportion* of object code uncertainty in the scene distribution ($H(O)$ or $H(O \mid F)$) resolved by knowledge of the representation.

*Remark* 2.19. Note that removing the scale effect of $H(O)$ or $H(O \mid F)$ does not remove other influences from the scene distribution, such as occlusion, background clutter, etc., which are often *the* points of comparison across datasets.

With binding information defined, we now show how it can be conveniently decoded by probes from model representations. Across variants above, there are two key terms: a representation-dependent term $H(O \mid Z)$ or $H(O \mid Z, F)$, and a prior term depending only on the data distribution, $H(O)$ or $H(O \mid F)$. We show how they can be calculated in Sections 2.2 and 2.3, respectively. And we summarize the full probing procedure in Section 2.4.

## 2.2. Probing Object Code on the Representation

We start with $H(O \mid Z)$ which is defined as

$$H(O \mid Z) = \mathbb{E}_{(z,o) \sim p(Z,O)} \left[ -\log p(o \mid z) \right].$$

A straightforward approach is to train a probe $q_\theta(o \mid z)$ to approximate $p(o \mid z)$. If we use cross-entropy loss

$$\mathcal{L}_{\mathrm{CE}}(\theta) = \mathbb{E}_{(z,o) \sim p(Z,O)} \left[ -\log q_\theta(o \mid z) \right], \quad (3)$$

we will be directly approximating $H(O \mid Z)$, provided that $q_\theta(o \mid z)$ is a good approximation of $p(o \mid z)$. The following theorem captures this intuition:

**Theorem 2.20** (Probe approximates binding uncertainty)**.** *Let $D(p(a \mid b) \parallel q(a \mid b))$ be the conditional relative entropy (Kullback–Leibler distance) between the two conditional probabilities $p(a \mid b)$ and $q(a \mid b)$, where $(a,b) \sim p(A,B)$. Then we have the following:*

$$\mathcal{L}_{\mathrm{CE}}(\theta) = H(O \mid Z) + D(p(o \mid z) \parallel q_\theta(o \mid z)).$$

*Proof in Appendix C.*

**Lemma 2.21.** *Since $D(p \parallel q_\theta) \geq 0$, we have*

$$\mathcal{L}_{\mathrm{CE}}(\theta) \geq H(O \mid Z).$$

*Remark* 2.22. Lemma 2.21 shows that the further we can minimize $\mathcal{L}_{\mathrm{CE}}(\theta)$, the closer we will be to ground truth $H(O \mid Z)$. This requires us to train the probes properly until convergence, with the appropriate training schedule, regularization (since we report cross-entropy loss on the test set), and particularly, a good model family to begin with. We therefore compare a range of probe families in our experiments, such as linear, quadratic, deep neural networks, and attention (more in Section 3).

A practical challenge in training a $q_\theta(o \mid z)$ probe is the exponential space of its labels $o$ (recall from Definition 2.5 that $O = o$ is a binary *vector*), which would require an exponential size dataset to cover the entire distribution space. We can circumvent this by decomposing the probe as follows:

$$q_\theta(o \mid z) = \prod_{k=1}^{K} q_\theta(o_k \mid o_{<k}, z), \quad (4)$$

where $o_k$ denotes the $k$-th component of the vector $o$, and $o_{<k}$ is the shorthand for $o_1, \cdots, o_{k-1}$.[5] This allows each $q_\theta(o_k \mid o_{<k}, z)$ to be trained individually with a single-dimension label for $o_k$ while incorporating $o_{<k}$ as prior.

---

[5] There is some abuse of notation here since earlier $o_i$ denotes objects (Definition 2.3). Here $o_k$ are the binary values that each $O_k$ can take, indicating whether the object is present or absent.

This amounts to probing for each object $o_k$ in the representation, with the "code" for the prior $o_{<k}$ objects given (Definitions 2.3, 2.5).[6] Their cross-entropy loss can then be summed to obtain the overall loss in Eq. (3):

$$\mathcal{L}_{\mathrm{CE}}(\theta) = \sum_{k=1}^{K} \mathbb{E}_{(z,o) \sim p(Z,O)} \left[ -\log q_\theta(o_k \mid o_{<k}, z) \right]. \quad (5)$$

To approximate $H(F \mid Z)$ in the conditional definition (Theorem 2.16), we can similarly use probes $q_\theta(f_k \mid f_{<k}, z)$ and sum their cross-entropy loss to obtain the $\mathcal{L}_{\mathrm{CE}}(\theta)$ for features, which will then be our probe estimate of $H(F \mid Z)$.

## 2.3. Estimating Dataset Priors

The prior terms in the binding definitions such as $H(O)$ and $H(F)$ can be empirically calculated using their entropy definition:

$$H(O) = \mathbb{E}_{o \sim p(O)} \left[ -\log p(o) \right],$$

where $p(O)$ is controllable for the synthetically generated datasets, similarly for $H(F)$.[7] Appendix F contains examples of calculating these quantities for our datasets.

## 2.4. Summary: Probing Binding on Model Representations

Finally, we summarize the procedure for probing binding information from model representations $Z$:

1. Find the dataset prior $H(O)$ (and $H(F)$ for conditional definition)

2. Train binding probes $q_\theta(o_k \mid o_{<k}, z)$ (and $q_\theta(f_k \mid f_{<k}, z)$ for conditional definition)

3. Sum their cross-entropy loss to obtain a probe estimate of $H(O \mid Z)$ (and $H(F \mid Z)$ for conditional definition)

4. Calculate binding information using Eq. (1) (Eq. (2) for conditional definition), with the option of normalization using Definition 2.17.

## 3. Results

Next, we probe ViT representations with real data. In particular, we are interested in understanding how binding works

---

[6] Note that a naive decomposition such as $q_\theta(o \mid z) = \prod_{k=1}^{K} q_\theta(o_k \mid z)$ is not true in general, since it does not account for the dependencies between $o_k$'s given $z$. However, we can make such assumptions *for our probe*. We perform a comparison between these two probe families in Appendix E.

[7] For natural datasets, since we do not directly control the distribution, we make particular assumptions on the data to make approximations on these quantities. Appendix F contains more details.

with different architectural components of the ViT (such as the `[CLS]` and the spatial tokens), and on different datasets with various challenges for binding.

In Sections 3.1 and 3.2, we ask which type of image representation in the ViT architecture is best for encoding binding: are summary tokens sufficient (e.g., the `[CLS]` or a global average pooling), or do we need the full set of spatial tokens? Both summary and spatial tokens are commonly used in pre-training objectives and downstream models: pre-training frameworks such as CLIP (Radford et al., 2021), simCLR (Chen et al., 2020), Barlow Twins (Zbontar et al., 2021), and other contrastive methods often use the `[CLS]` or globally pooled representations as the input to their training objectives; Vision-Language Models (VLMs), on the other hand, utilize the full set of spatial tokens in their cross-modality processing with language (Liu et al., 2023; Bai et al., 2023). Studying how well binding is encoded in these different types of representations can thus guide better model development and evaluation.

In Section 3.3, we study the data aspects of binding. We probe on datasets with different challenges for binding: datasets with an increasing number of features and objects in the space, datasets with different levels of occlusion, and natural datasets with high-level, semantic features.

### 3.1. To what extent does the `[CLS]` summary token encode binding?

To test binding, we create a synthetic ColorShape dataset with 8 colors and 8 shapes in the feature space, and 64 objects in the object space, each taking one shape and one color. The resulting feature and object codes have $\dim(F) = 16, \dim(O) = 64$. To create a balanced dataset for both the feature and object codes, we sample each image as follows: randomly choose 6 colors and 6 shapes, and out of the 36 possible objects, choose 18 objects randomly that cover all 6 colors and 6 shapes chosen, and place them in the image at random locations without overlap. The resulting baseline accuracy is $6/8 = 75\%$ for feature probes $q_\theta(f_k \mid f_{<k}, z)$ and $1 - 18/64 = 71.9\%$ for object probes $q_\theta(o_k \mid o_{<k}, z)$, where the baseline is a trivial prediction of a constant 1 and 0, respectively.[8] We report error reduction, (accuracy – baseline) / (1 – baseline) $\times 100\%$, in lieu of raw accuracy for our probes.

To test whether our probes learn generalizable patterns instead of memorization, we split the dataset into training, validation, and test sets. They each contain a disjoint set of feature codes $F$ and object codes $O$. The latter occurs naturally by virtue of the large combinatorial space of the

object codes ($\sim 10^{12}$), so every image almost certainly has a different object code. The feature code split is implemented separately. The resulting training set contains 39,200 samples, validation 9,800 samples, and test 9,800 samples.

We then compute the dataset entropies $H(F)$ and $H(O)$ for each split. Since all feature and object codes have equal probability in our setup, we only need the number of all possible feature codes and object codes under the feature code constraint for each split, and then take their logarithm to find the entropies. Both can be calculated with basic combinatorics, which we detail in Appendix F. On the test set for which we report binding information metrics, $H(O) = 39.9$ bits and $H(F) = 7.0$ bits.

Using this dataset, we probe for binding information from the `[CLS]` token of the DINOv2-Large ViT (Oquab et al., 2024). We train autoregressive probes $q_\theta(o_k \mid o_{<k}, z)$ to approximate the $H(O \mid Z)$ term in the binding definition (one for each $o_k$, so 49 in total). To incorporate the conditional prior $o_{<k}$, we directly concatenate $o_{<k}$ with representation $z$: $x = [z || o_{<k}]$.[9] We experiment with three probe families:

- Linear probes: $\ell_k(x) = W_k x + b_k$

- Quadratic probes: $\ell_k(x) = x^\top W_k x + b_k$

- Deep neural network (DNN): $\ell_k(x) = f_{\mathrm{DNN}}(x) + b_k$ where $f_{\mathrm{DNN}}$ is 4 layers of 1024 width with GELU activations.

where $q_\theta(o_k = 1 \mid o_{<k}, z) = \sigma(\ell_k(x))$.

We show the object code and feature code probing results in Table 1. All three probe families generalize well to unseen object and feature codes. In terms of accuracy, feature decoding can reach $90\%$ in error reduction, while object decoding reaches merely around $65\%$ percent. Linear probes perform the worst in both cases. The DNN probe performs the best, although only slightly better than the much smaller quadratic probe. Given that DNNs are known as "universal approximators" (Hornik et al., 1989), the DNN probes (with 3M parameters) reflect our best attempt at approaching the true binding (i.e., object) information in the representation, by closing the gap between the probe model $q_\theta(o \mid z)$ and the true distribution $p(o \mid z)$ (Lemma 2.21). Since the quadratic probes achieve only slightly higher loss than the DNN probes, it is fair to assume that *most* binding (i.e., object) information and feature information in the representation is quadratic, and that higher-order interactions add little to no extra information decodable.[10]

---

[8]Additionally, feature information has only minor leakage to objects: a baseline model that assumes all conjunctions of present features exist as objects will have accuracy $[18 + (64 - 36)]/64 = 71.9\%$.

[9]There are, of course, other ways to incorporate the $o_{<k}$ prior in our probe. We ablate on conditioning on $o_{<k}$ in Appendix E.

[10]That is, of course, assuming that the DNN probe indeed approaches the upper-bound of the binding information.

*Table 1.* **Average performance of object and feature code probes for different probe families when** $z$ **is the** `[CLS]` **token.** We report error reduction (ER) relative to the trivial majority-label baseline, which achieves $71.9\%$ accuracy for object codes and $75.0\%$ accuracy for feature codes. Numbers in parentheses denote the number of parameters relative to the corresponding linear probe.

| Probe type | Probe family | Train loss ↓ / ER ↑ | Val loss ↓ / ER ↑ | Test loss ↓ / ER ↑ |
|---|---|---|---|---|
| $q_\theta(o_k \mid o_{<k}, z)$ | Linear ($\times 1$) | $33.8 / 38.4\%$ | $34.3 / 37.4\%$ | $34.2 / 37.4\%$ |
| | Quadratic ($\times 64$) | $21.4 / 64.4\%$ | $22.0 / 63.3\%$ | $22.0 / 63.0\%$ |
| | DNN ($\times 2953$) | $\mathbf{19.9} / \mathbf{65.8}\%$ | $\mathbf{20.6} / \mathbf{64.4}\%$ | $\mathbf{20.6} / \mathbf{64.4}\%$ |
| $q_\theta(f_k \mid f_{<k}, z)$ | Linear ($\times 1$) | $4.2 / 73.2\%$ | $4.5 / 69.6\%$ | $4.4 / 70.4\%$ |
| | Quadratic ($\times 64$) | $1.5 / 90.8\%$ | $1.6 / 89.2\%$ | $1.6 / 89.6\%$ |
| | DNN ($\times 3042$) | $\mathbf{1.1} / \mathbf{92.4}\%$ | $\mathbf{1.3} / \mathbf{90.8}\%$ | $\mathbf{1.3} / \mathbf{91.2}\%$ |

To further understand how this quadratic binding information may be structured, we then devise a version of the quadratic probe where we reuse parameters across $o_k$ probes when the objects $o_k$ *share features* (e.g., red square probe and red circle probe share the "red" portion of their weights). More formally, we set $W_k = U_{\text{color}}^\top V_{\text{shape}}$ and the probe becomes

$$\ell_k(x) = x^\top U_{\text{color}}^\top V_{\text{shape}} x + b_k,$$

where $U_{\text{color}}$ is shared for all objects with the same color, and $V_{\text{shape}}$ is shared for all objects with the same shape. From Table 2, we observe a small 2.4-bit increase in loss relative to the non-parameter-reusing $o_k$ probes. This suggests the strong presence of quadratic binding (i.e., object) information in the `[CLS]` token *as the dot product between color and shape projections*, a conjunctive encoding of objects based on features.

*Table 2.* **Comparison of quadratic object code probes with and without parameter reuse.** We report error reduction (ER) relative to the trivial majority-label baseline, which achieves $71.9\%$ accuracy for object codes. Numbers in parentheses denote the number of parameters relative to the linear probe.

| Parameter reuse | Train loss ↓ / ER ↑ | Val loss ↓ / ER ↑ | Test loss ↓ / ER ↑ |
|---|---|---|---|
| Yes ($\times 1/6$) | $23.9 / 58.7\%$ | $24.4 / 58.0\%$ | $24.4 / 57.7\%$ |
| No ($\times 1$) | $21.4 / 64.4\%$ | $22.0 / 63.3\%$ | $22.0 / 63.0\%$ |
| $\Delta$ (Yes − No) | $+2.5 / -5.7$ pp | $+2.4 / -5.3$ pp | $+2.4 / -5.3$ pp |

Next we compute the amount of binding information and proportion of dataset binding uncertainty resolved, with or without conditioning on features, as we have defined in Section 2.1. Table 3 shows the results on the test set, using the probe losses tabled above. To be fair across probe types for the feature conditioned results, we use the quadratic feature probe loss for all of their feature terms, varying only the object probe type. The DNN probe, again, decodes the most amount of binding information, and only slightly more than the quadratic probe. Assuming that the DNN probe approaches the true binding information in the representation, our results show that the `[CLS]` token encodes less than half ($48.5\%$) of the binding information. If we wish to separate binding information from feature information,

the `[CLS]` encodes only $42.4\%$ of the binding information beyond what can be explained by features.

*Table 3.* **Probe-estimated binding information** $\mathbf{B}_\mathcal{O}(Z)$**, conditional binding information** $\mathbf{B}_{\mathcal{O},\mathcal{F}}^*(Z)$**, and their normalized forms** $\boldsymbol{\beta}_\mathcal{O}(Z)$ **and** $\boldsymbol{\beta}_{\mathcal{O},\mathcal{F}}^*(Z)$**,** recorded on the test set. $H(O) = 39.9$ bits and $H(F) = 7.0$ bits on this test set. All conditional results use the loss of the quadratic feature probe for the $H(F \mid Z)$ term, varying the object code probe type.

| Probe type | $\mathbf{B}_\mathcal{O}(Z)$ (bits) ↑ | $\boldsymbol{\beta}_\mathcal{O}(Z)$ ↑ | $\mathbf{B}_{\mathcal{O},\mathcal{F}}^*(Z)$ (bits) ↑ | $\boldsymbol{\beta}_{\mathcal{O},\mathcal{F}}^*(Z)$ ↑ |
|---|---|---|---|---|
| Linear | $5.7$ | $14.3\%$ | $0.3$ | $0.8\%$ |
| Quadratic | $17.9$ | $44.9\%$ | $12.5$ | $37.9\%$ |
| DNN | $\mathbf{19.4}$ | $\mathbf{48.5}\%$ | $\mathbf{13.9}$ | $\mathbf{42.4}\%$ |

### 3.2. To what extent does the full set of spatial tokens encode binding?

If a single summary token performs poorly for binding, would the full set of spatial tokens retain more binding information? Again, we use the same ColorShape dataset and DINOv2-Large ViT to probe for binding. Directly concatenating all spatial tokens, however, would produce a very high-dimensional input to the probe, making the number of parameters explode. So instead, we use a simplified attention probe that queries on the spatial tokens and learns their dynamically weighted mean:

Let $z = \{s_i\}_{i=1}^N$ denote the full set of final-layer spatial tokens of the ViT. Each probe $q_\theta(o_k \mid o_{<k}, z)$ learns a query vector $q_k$ conditioned on $o_{<k}$:

$$q_k = g_k(o_{<k}).$$

The query scores each spatial token by a dot product,

$$a_{k,i} = \frac{\exp(q_k^\top s_i)}{\sum_{j=1}^N \exp(q_k^\top s_j)},$$

and forms a weighted spatial representation

$$\bar{s}_k = \sum_{i=1}^N a_{k,i} s_i.$$

A final quadratic readout layer then predicts

$$q_\theta(o_k = 1 \mid o_{<k}, \{s_i\}_{i=1}^N) = \sigma(s_k^\top W_k \bar{s}_k + b_k).$$

This probe is simpler than a typical transformer attention layer in that it uses learned queries, but no learned key or value projections. As shown in Table 4, this simplified attention probe can perform at around $97\%$ in error reduction for object decoding, far outperforming the best DNN probe on the `[CLS]` token only.

*Table 4.* **Average performance of attention probes with quadratic readout using the full set of spatial tokens**, compared to the best-performing [CLS] probe for object and feature code prediction. We report error reduction (ER) relative to the trivial majority-label baselines, which achieve 71.9% accuracy for object codes and 75.0% accuracy for feature codes.

| Probe type | Probe family | Train loss ↓ / ER ↑ | Val loss ↓ / ER ↑ | Test loss ↓ / ER ↑ |
|---|---|---|---|---|
| $q_\theta(o_k \mid o_{<k}, z)$ | Attention + spatial | **2.9 / 97.2**% | **3.2 / 96.8**% | **3.1 / 96.8**% |
|  | DNN + [CLS] | 19.9 / 65.8% | 20.6 / 64.4% | 20.6 / 64.4% |
| $q_\theta(f_k \mid f_{<k}, z)$ | Attention + spatial | **1.0 / 92.8**% | **1.2 / 91.2**% | **1.2 / 91.6**% |
|  | DNN + [CLS] | 1.1 / 92.4% | 1.3 / 90.8% | 1.3 / 91.2% |

The binding information metrics calculated from these probes are listed in Table 5. As can be seen, the attention probe on the full set of spatial tokens decodes 92.2% of binding information, and for the binding information beyond what can be explained by features, there is 94.1% decoded. Both significantly outperform the best DNN probe on the [CLS] token only.

*Table 5.* **Probe-estimated binding information** $\mathbf{B}_\mathcal{O}(Z)$**, conditional binding information** $\mathbf{B}^*_{\mathcal{O},\mathcal{F}}(Z)$**, proportion of dataset binding information resolved** $\boldsymbol{\beta}_\mathcal{O}(Z)$**, and their normalized forms** $\boldsymbol{\beta}_\mathcal{O}(Z)$ **and** $\boldsymbol{\beta}^*_{\mathcal{O},\mathcal{F}}(Z)$**, recorded on the test set.** $H(O) = 39.9$ bits and $H(F) = 7.0$ bits on this test set. Attention probe's conditional results use the loss of the attention feature probe, while DNN's conditional results use the loss of the quadratic feature probe, as in Table 3.

| Probe type | $\mathbf{B}_\mathcal{O}(Z)$ (bits) ↑ | $\boldsymbol{\beta}_\mathcal{O}(Z)$ ↑ | $\mathbf{B}^*_{\mathcal{O},\mathcal{F}}(Z)$ (bits) ↑ | $\boldsymbol{\beta}^*_{\mathcal{O},\mathcal{F}}(Z)$ ↑ |
|---|---|---|---|---|
| Attention + spatial | **36.8** | **92.2**% | **31.0** | **94.1**% |
| DNN + [CLS] | 19.4 | 48.5% | 13.9 | 42.4% |

Here in this probe, the attention weights $a_{k,i}$ can be interpreted as selecting which spatial tokens are most useful for decoding $o_k$. Qualitatively, we find that this routing process is highly accurate in our probes: when the target object $o_k$ exists in the image (with label $o_k = 1$), the highest attention score $a_{k,i}$ almost always maps to the target object patch.

### 3.3. To what extent can binding be learned on different datasets?

Does our binding framework capture the intrinsic difficulty of binding in the data distribution? We measure binding information on the following datasets with different binding challenges:

- **ColorShape** with the number of colors and shapes varying from 1 to 7 each, where the growing feature (and object) space can make it increasingly difficult for the model to bind well. We simplify the data distribution for easier comparison.

- **CLEVR** with varying degree of occlusion between objects (Johnson et al., 2016), where occlusion can lead to ambiguous boundaries between objects.

- **Visual Genome** with densely annotated natural features (Krishna et al., 2017) where feature and object learning may be intrinsically more difficult.

Appendix G contains the detailed setups for each dataset. We note here that we construct image representations by concatenating the [CLS] token with the mean-pooled spatial tokens from the final layer of the ViT.

**Influence of feature and object space complexity on binding.** We run the CLIP ViT-L/14 (224px) model on the ColorShape dataset while varying the number of colors and shapes from 1 to 7. Figure 3 shows the results for all 49 configurations on this dataset. As shown in the figure (right), as the number of feature values (e.g., colors and shapes) increases, the space of possible bindings grows exponentially in the number of feature combinations, leading to a rapid increase in binding uncertainty (e.g., from 1 to $2^{43}$ possibilities as the numbers of colors and shapes grow from 1 to 7). While the fraction of binding information captured by the model decreases with increasing complexity, it does not decay exponentially (Figure 3 Middle).

**Influence of occlusion on binding.** We probe the representations of DINOv2-Large on datasets with different levels of occlusion (examples shown in Figure 5), where we vary the level of occlusion (by adjusting the camera elevation and angle of the scene) As shown in Table 6, occlusion has a notable influence on a model's ability to bind: binding information decodable monotonically decreases with higher levels of occlusion.

*Table 6.* **Effect of occlusion on binding for DINOv2-Large**, using CLEVR dataset with 4 colors and 3 shapes. Camera elevation serves as a proxy for **occlusion level** (higher means less occlusion). $\Delta\beta(Z)$ is measured relative to the most occluded setting (0.6).

| $ht.$(Camera) | $\mathbf{B}^*_{\mathcal{O},\mathcal{F}}(Z)$ (bits) | $\boldsymbol{\beta}^*_{\mathcal{O},\mathcal{F}}(Z)$ | $\Delta\boldsymbol{\beta}^*_{\mathcal{O},\mathcal{F}}(Z)$ (pp) |
|---|---|---|---|
| 0.6 | 5.8 | 45.0% | 0.0 |
| 1.2 | 6.6 | 51.1% | +6.1 |
| 1.8 | 6.9 | 53.3% | +8.3 |
| 2.5 | 7.1 | 55.4% | +10.4 |
| 3.2 | 7.6 | 58.7% | +13.7 |

**Binding on natural datasets.** Table 7 shows binding information on the natural Visual Genome dataset, along with previous synthetic datasets across three models. Although natural features and objects are more difficult to learn (often emerging in later layers of a neural network), the models achieve comparable levels of binding with synthetic datasets. Additionally, we note from the table that our binding measure also captures model capability: increasing CLIP's input resolution from 224px to 336px improves binding on all datasets, suggesting that finer spatial representations better support object-feature binding.

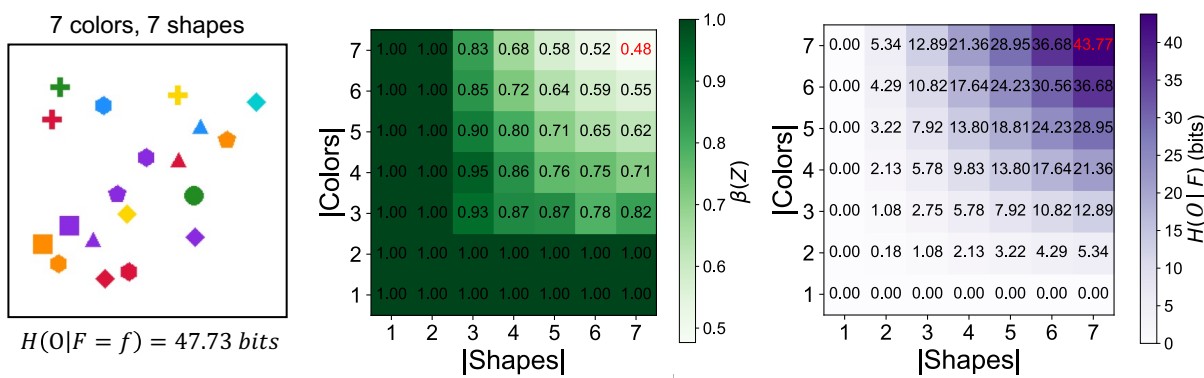

*Figure 3.* **Binding degrades with increasing feature space complexity in ColorShape. Middle:** We run the CLIP ViT-L/14 (224px) model on the Color-Shape dataset with all 49 configurations of numbers of colors and shapes, and report the normalized binding information $\beta^*_{\mathcal{O},\mathcal{F}}(Z)$. The percentage of binding information captured by the model decreases as the dataset becomes more complex, involving more objects with more colors and shapes. **Left:** With the full feature code $F$ set to 7 colors and 7 shapes (all present), the uncertainty of binding is captured by $H(O \mid F = f)$ which is 47.73 bits. **Right:** The binding uncertainty of the dataset distribution $H(O|F)$ for each configuration of numbers of colors and shapes.

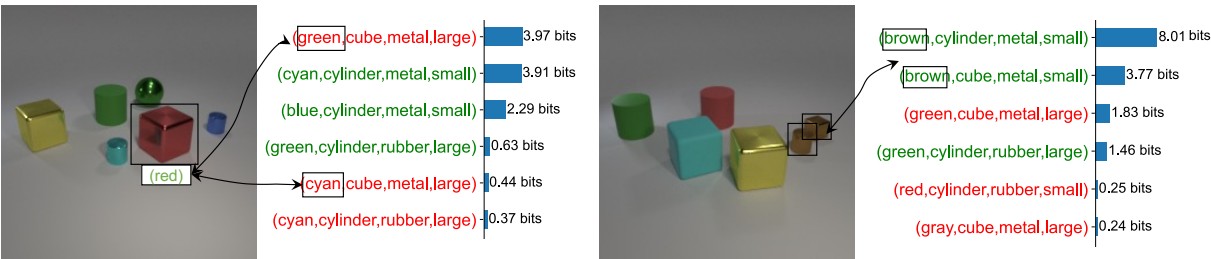

*Figure 4.* **CLEVR examples of object-code uncertainty revealed by binding probes.** Blue bars ($H(O \mid Z)$) indicate uncertainty in predicting objects. **Left:** Color misattribution in binding. **Right:** Uncertainty on the joint presence of a brown cylinder and a brown cube.

*Table 7.* **Conditional binding information $\beta^*_{\mathcal{O},\mathcal{F}}(Z)$ for different models and datasets.** The ColorShape dataset here has 7 colors and 7 shapes, with feature and object code distributions detailed in Appendix G. VG:Color and VG:TopAttr are two subsets of Visual Genome.

| Model | ColorShape | CLEVR | VG:COLOR | VG:TopAttr |
|---|---|---|---|---|
| DINOv2-Large | 41.8% | 61.0% | 41.8% | **47.0**% |
| CLIP ViT-L/14 (224px) | 47.7% | 68.1% | 45.8% | 39.9% |
| CLIP ViT-L/14 (336px) | **56**.4% | **68.5**% | **46.4**% | 45.6% |

## Discussion

In our work, we defined binding with an information-theoretic framework and devised probing methods for measuring binding in model representations. We believe there are many benefits for taking a representational approach to defining binding: first, virtually all models are representation learners. Our definition can thus be widely applied for measurement and for comparison. Second, compared to directly measuring accuracy in binding tasks, the information approach (measuring entropy) takes into account the decision uncertainty of the model. Third, a representation measurement can enable us to understand the structure of

the information by using differentially structured probes, as we have shown with the quadratic binding structure of the `[CLS]`. Finally, a representation-level diagnosis can guide approaches in model pre-training that specifically target better representations, such as the self-supervised pre-training of world models (Assran et al., 2025).

**Future work** can focus on improving binding performance of the latest models using the binding probe performances as objectives, and exploring potential architectural inductive biases to improve binding.

Our work does have **limitations**. Our framework relies on predefined discrete feature vocabularies; future work could extend it to continuous features with continuous probes. In addition, our probes measure decodable binding information rather than whether the model causally uses such information during downstream inference, so high probe performance may reflect latent information inaccessible to the model's native readout mechanisms. Finally, the conditional definition ($B^*_{O,F}(Z)$) also assumes that feature code ($F$) can be reliably inferred from object code ($O$), which may not hold in noisy biological or human perception.

## Acknowledgements

The authors would like to thank the anonymous reviewers for their helpful feedback.

## Impact Statement

This paper presents work whose goal is to advance the field of machine learning and neuroscience. There are many potential societal consequences of our work, including those related to improved visual capabilities of artificial intelligence and better understanding of biological vision.

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

# A. Related Work

**The Binding Problem**  With its roots in early artificial intelligence (Rosenblatt, 1962), the binding problem has puzzled computer scientists and cognitive neuroscientists for over half a century (Greff et al., 2020; Von der Malsburg, 1999; Feldman, 2013; Roskies, 1999; Treisman, 1996). Since early artificial vision models were in their infancy and lacked the capacity for complex scene decomposition, the initial momentum for understanding feature binding came largely from psychology and neuroscience (Treisman, 1999; von der Malsburg, 1981). Although there is yet no consensus on the exact neural mechanisms of binding in the brain (Robertson, 2003; Wolfe, 2012; Di Lollo, 2012; Scholte & de Haan, 2025; Roelfsema & Serre, 2025), significant progress has been made in identifying some of its core mechanistic components (Treisman & Gelade, 1980; Reynolds & Desimone, 1999; Roelfsema, 2023; Singer & Gray, 1995). These biological insights have inspired architectures designed to explicitly address binding, whether through temporal synchrony (Miyato et al., 2024; Gopalakrishnan et al., 2024) or attention-based mechanisms (Salehi et al., 2024).

**Feature Binding in Deep Neural Networks**  Following the unprecedented success of deep learning in computer vision (Krizhevsky et al., 2012; LeCun et al., 2015), the binding problem has gained renewed interest within the machine learning community. Earlier approaches attempted to address binding through explicit object-centric architectural and training biases (Locatello et al., 2020; Puebla & Bowers, 2025; Assouel et al., 2025a; Aydemir et al., 2023), known as Object-Centric Learning (OCL) (Locatello et al., 2020; Greff et al., 2019). Approaches such as Slot Attention (Locatello et al., 2020) and MONet (Burgess et al., 2019) decompose inputs to permutation-invariant "slots" via iterative refinement, each representing an object. Although these models excel at grouping spatially coherent regions (object-part binding), they arguably fall short in feature binding, often struggling to correctly associate complex attributes like texture with their respective objects (Karazija et al., 2021). Also, their general applicability is largely surpassed by the performance and scalability of self-supervised Vision Transformers (ViTs) (Siméoni et al., 2025; He et al., 2022; Zhou et al., 2021) and multi-modal models (Radford et al., 2021; Girdhar et al., 2023; Rubinstein et al., 2025). State-of-the-art ViTs now excel in both global recognition and dense prediction tasks (e.g., segmentation) (Siméoni et al., 2025) without requiring explicit architectural components or training process for binding. This has prompted a growing body of research investigating the degree to which binding capabilities naturally emerge in these models (Li et al., 2025), identifying the underlying mechanisms that drive their partial success (Okawa et al., 2023; Seitzer et al., 2022; Wu et al., 2025), and proposing methods to remedy their limitations (Seo et al., 2025; Hu et al., 2024; Koishigarina et al., 2025).

**Probing for Binding**  A rigorous evaluation of feature binding requires both a suitably complex dataset and a precise metric for quantification. An ideal dataset must include scenes with multiple objects to challenge the model's binding capability, while providing ground-truth annotations that link objects to their specific attributes. Standard benchmarks such as MS-COCO (Lin et al., 2015) and synthetic environments like CLEVR (Johnson et al., 2016) have traditionally served this role, while recent works have introduced specialized datasets to target fine-grained compositional understanding (Liu et al., 2024). In terms of measurement, existing approaches fall into two categories. The "symptomatic" approaches assess binding indirectly via performance on high-level downstream tasks, such as visual reasoning (Campbell et al., 2025), text-image retrieval errors (Yuksekgonul et al., 2023) and compositional generalization (Lewis et al., 2024; Pearson et al., 2025). Conversely, "mechanistic" approaches aim to evaluate binding directly within the internal representations, offering a task-agnostic measure of how information is organized and bound at the low level (Li et al., 2025).

Recent studies have shown that vision-language models (e.g., CLIP) behave like bag-of-words and do not reliably bind the correct attributes to their corresponding entity (Koishigarina et al., 2025; Yuksekgonul et al., 2023), but these analyses rely primarily on linear probes. Since binding represents a combinatorial problem where linear separation may require exponentially high dimensionality, we argue that linear probes are insufficient. We therefore train non-linear probes to capture these complex relationships. (Li et al., 2025) propose quadratic probes to evaluate the object-binding property of *IsSameObject*. However, this approach is not easily generalizable and implicitly assumes that each patch corresponds to a single, well-isolated object. Here, we provide a more formalized framework for binding that, in principle, extends to arbitrary models, representations, and feature definitions.

## B. Data Processing Inequality

**Theorem B.1** (Internal processing does not increase binding information). *A model typically has several layers, yielding internal representations $Z_1, \cdots, Z_n$. For a deterministic model where*

$$Z_1 = g_0(X), \quad Z_{i+1} = g_i(Z_i), \quad i \in [1 \ldots n],$$

*there is*

$$\mathbf{B}_{\mathcal{O}}(X) \geq \mathbf{B}_{\mathcal{O}}(Z_1) \geq \cdots \geq \mathbf{B}_{\mathcal{O}}(Z_n),$$

*where we define $\mathbf{B}_{\mathcal{O}}(X) := I(O; X \mid F)$. That is, any representation $Z_i$ cannot increase binding information in the input $X$, and neither does binding information increase through the internal processing of the model.*

*Proof.* Since $Z_{i+1} \perp\!\!\!\perp Z_{<i} \mid Z_i$, we have a Markov Chain: $Z_1 \to \cdots \to Z_n$. Since $X$ is generated from $p(X \mid O)$ (where $O$ specifies the feature values of objects), we can add $O$ to the Markov Chain: $O \to X \to Z_1 \to \cdots \to Z_n$. For simplicity of notation, we denote $X := Z_0$.

For any $0 \leq i < j$, we first show that $I(Z_j; O \mid Z_i, F) = 0$. $I(Z_j; O \mid Z_i, F) = H(Z_j \mid Z_i, F) - H(Z_j \mid O, Z_i, F)$. Since $0 \leq H(Z_j \mid Z_i, F) \leq H(Z_j \mid Z_i) = 0, 0 \leq H(Z_j \mid O, Z_i, F) \leq H(Z_j \mid Z_i) = 0$, we have $H(Z_j \mid Z_i, F) = H(Z_j \mid O, Z_i, F) = 0$. Hence $I(Z_j; O \mid Z_i, F) = 0$.

By the chain rule of information,

$$I(Z_i; O \mid F) = I(Z_i; O \mid F) + I(Z_j; O \mid F, Z_i) = I(Z_i, Z_j; O \mid F) = I(Z_j; O \mid F) + I(Z_i; O \mid F, Z_j) \geq I(Z_j; O \mid F).$$

$\square$

## C. Proofs

### C.1. Theorem 2.14

*Proof.* We know that

$$I(O; Z) = H(O) - H(O \mid Z)$$
$$I(F; Z) = H(F) - H(F \mid Z) \tag{6}$$
$$\mathbf{B}_{\mathcal{O}}(Z) = H(O \mid F) - H(O \mid F, Z) \tag{7}$$

Adding Eqs. 6 and 7, we have $I(F; Z) + \mathbf{B}_{\mathcal{O}}(Z) = H(O, F) - H(O, F \mid Z) = H(F) - H(O \mid Z) = I(O; Z)$. The penultimate equality is due to the fact that $F$ is a deterministic function of $O$. $\square$

### C.2. Theorem 2.16

*Proof.* By definition,

$$\mathbf{B}^*_{\mathcal{O}, \mathcal{F}}(Z) = I(O; Z \mid F) = H(O \mid F) - H(O \mid Z, F).$$

Since $F$ is a deterministic function of $O$, $H(F \mid O) = H(F \mid O, Z) = 0$. Hence,

$$H(O \mid F) = H(O, F) - H(F) = H(O) - H(F),$$

and

$$H(O \mid Z, F) = H(O, F \mid Z) - H(F \mid Z) = H(O \mid Z) - H(F \mid Z).$$

Substituting gives

$$\mathbf{B}^*_{\mathcal{O}, \mathcal{F}}(Z) = H(O) - H(O \mid Z) - H(F) + H(F \mid Z).$$

$\square$

## C.3. Theorem 2.20

*Proof.* By definition,

$$\mathcal{L}_{\text{CE}}(\theta) = \mathbb{E}_{(o,z)\sim p(O,Z)}\left[-\log q_\theta(o \mid z)\right].$$

Adding and subtracting $\log p(o \mid z)$, we get

$$\mathcal{L}_{\text{CE}}(\theta) = \mathbb{E}_{(o,z)}\left[-\log p(o \mid z)\right] + \mathbb{E}_{(o,z)}\left[\log \frac{p(o \mid z)}{q_\theta(o \mid z)}\right].$$

The first term is $H(O \mid Z)$, and the second term is

$$D(p(o \mid z) \parallel q_\theta(o \mid z)).$$

Therefore,

$$\mathcal{L}_{\text{CE}}(\theta) = H(O \mid Z) + D(p(o \mid z) \parallel q_\theta(o \mid z)).$$

$\square$

# D. Experimental Setup

We train probes on frozen visual representations. A summary of the default probe architectures and training hyperparameters is provided in Table 8. All experiments are run on NVIDIA RTX 4090 GPUs.

In the dataset comparisons, we construct image representations by concatenating the `[CLS]` token with the mean-pooled spatial tokens from the final layer of the vision encoder.

*Table 8.* **Default probe architectures and training hyperparameters.**

| Trainer component | Setting |
| --- | --- |
| Batch size | 512 |
| Gradient Accumulation | 2 |
| Learning rate | $1 \times 10^{-3}$ |
| Weight decay | $1 \times 10^{-4}$ |
| Epochs | 40 epochs (converged for all probes) |
| LR scheduler | StepLR (step size = 8, $\gamma = 0.2$) |

**Models.** We evaluate feature binding behavior using a diverse set of pretrained Vision Transformer backbones (Table 9). All models are used in a frozen setting.

*Table 9.* **Vision Transformer backbones used for evaluation.**

| Model family | HuggingFace identifier |
| --- | --- |
| DINOv2-Large | `facebook/dinov2-large` |
| CLIP ViT-L/14 (224px) | `openai/clip-vit-large-patch14` |
| CLIP ViT-L/14 (336px) | `openai/clip-vit-large-patch14-336` |

# E. Ablation on $o_{<k}$ labels

Table 10 shows that ablating on the conditionals leads to a notable decrease in binding information decoded, suggesting that it is necessary to condition on the $o_{<k}$ labels as required by the decomposition in Eq. 4.

*Table 10.* **Ablation of conditioning quadratic object code probes on previous object codes** $o_{<k}$. We report error reduction (ER) relative to the trivial majority-label baseline, which achieves 71.9% accuracy for object codes.

| Condition on $o_{<k}$ | Train loss ↓ / ER ↑ | Val loss ↓ / ER ↑ | Test loss ↓ / ER ↑ |
|---|---|---|---|
| Yes | **21.4 / 64.4**% | **22.0 / 63.3**% | **22.0 / 63.0**% |
| No | 26.6 / 52.3% | 27.3 / 50.9% | 27.3 / 50.9% |
| $\Delta$ (Yes – No) | $-5.2$ / $+12.1$ pp | $-5.3$ / $+12.5$ pp | $-5.3$ / $+12.1$ pp |

## F. Estimating Dataset Priors

**Computing $H(O)$ and $H(F)$ for ColorShape (8 colors and 8 shapes)**  To find $H(O)$ and $H(F)$ for the ColorShape dataset, we only need to find the size of their support in each split and then take their logarithm, since they are both evenly distributed. Prior to splitting, there are $\binom{8}{6} \times \binom{8}{6} = 784$ possibilities of choosing $F$. The training/validation/test sets each takes 522/131/131 disjoint values of $F$. Hence $H_{\text{train}}(F) = \log_2(522) = 9.0$ bits, $H_{\text{val}}(F) = H_{\text{test}}(F) = \log_2(131) = 7.0$ bits.

Next we find the support size of $O$ for each split. For each *chosen* set of 6 colors and 6 shapes, we now find the number of ways to choose 18 objects out of the 36 possibilities while ensuring coverage of all features. We use the inclusive-exclusion formula by first choosing 18 objects out of 36, $\binom{36}{18}$, and then subtracting that by scenarios where we miss at least one feature $-\binom{6}{1}\binom{5\times 6}{18} - \binom{6}{1}\binom{6\times 5}{18}$ and adding back scenarios where we miss at least two features $\binom{6}{2}\binom{4\times 6}{18} + \binom{6}{1}\binom{6}{1}\binom{6\times 4}{18} + \binom{6}{2}\binom{5\times 5}{18}$, and so on. More compactly, the support size of $O$, given a *chosen* set of 6 colors and 6 shapes, is as follows:

$$\sum_{i=0}^{6}\sum_{j=0}^{6}(-1)^{i+j}\binom{6}{i}\binom{6}{j}\binom{(6-i)(6-j)}{18} = 8,058,525,440.$$

Now, $O$ is different for different chosen sets of colors and shapes, so the total support size is the above multiplied by the support size of $F$ in each split. Hence $H_{\text{train}}(O) = \log_2(8,058,525,440 \times 522) = 41.9$ bits, and similarly $H_{\text{val}}(O) = H_{\text{test}}(O) = \log_2(8,058,525,440 \times 131) = 39.9$ bits.

**Computing $H(O \mid F)$ for ColorShape ($1-7$ colors/shapes) and CLEVR**  In ColorShape and CLEVR, we generate synthetic data with full control, enforcing a uniform distribution over all valid binding configurations, with the number of objects capped at $k_{\max}$. Under this assumption,

$$H(O \mid F = f) = \log_2\left(\sum_{k \in \mathcal{K}(f)} N_k(a_1, \ldots, a_G)\right),$$

where $f$ denotes a fixed feature realization, $\mathcal{K}(f)$ is the set of admissible object counts consistent with $f$, and $N_k(a_1, \ldots, a_G)$ counts the number of valid binding configurations with exactly $k$ objects.

Because the distribution of $F$ is also known and tractable in these datasets, we can compute the true conditional binding entropy $H(O \mid F)$ exactly by averaging $H(O \mid F = f)$ over feature realizations.

**Computing $H(O \mid F)$ for Visual Genome.**  We again approximate $H(O \mid F)$ by enumerating binding configurations consistent with $F = f$. Here, however, we cannot control for $O$ being uniform, so we need to use the empirical dataset distribution of $O$ as an approximation. We assume that each object instance occurs independently: this means the probability of a binding $O = o$ is the product of probabilities of each object instance in that binding assignment. The probability of each object instance (e.g. red car, blue balloon) is easily computed over the dataset. This then gives $H(O \mid F = f)$ as well as $p(f)$, and after averaging over $f \sim p(f)$, yields $H(O \mid F)$.

## G. Details of datasets in Section 3.3

**ColorShape ($1-7$ colors/shapes)**  We consider a ColorShape dataset with 7 colors and 7 shapes. The maximum number of objects is capped at $\lfloor N_1 N_2 / 2 \rfloor$. We evaluate the model across all possible numbers of color–shape pairs. The number of training samples is scaled proportionally to $N_1 \cdot N_2$. The dataset consists of 48,000 samples.

Different from the ColorShape with 8 colors and 8 shapes, and to simplify calculation, we generate images $X \sim P(X \mid O)$ with $O$ *uniform* over its allowed values in its space $\Omega_O$, i.e., every multi-object scene, determined by the object types in the scene, is equally likely.

**CLEVR**   To get progressively closer to real-world scenarios that include more feature types, we evaluate binding on CLEVR (Johnson et al., 2016), for which we generate images with 3-D objects out of 8 colors, 3 shapes, 2 materials, and 2 sizes, with a total of 96 possible feature combinations. We illustrate the different occlusion levels of the CLEVR dataset in Figure 5, which are controlled by camera elevation and pitch angle. The dataset consists of 24,000 samples.

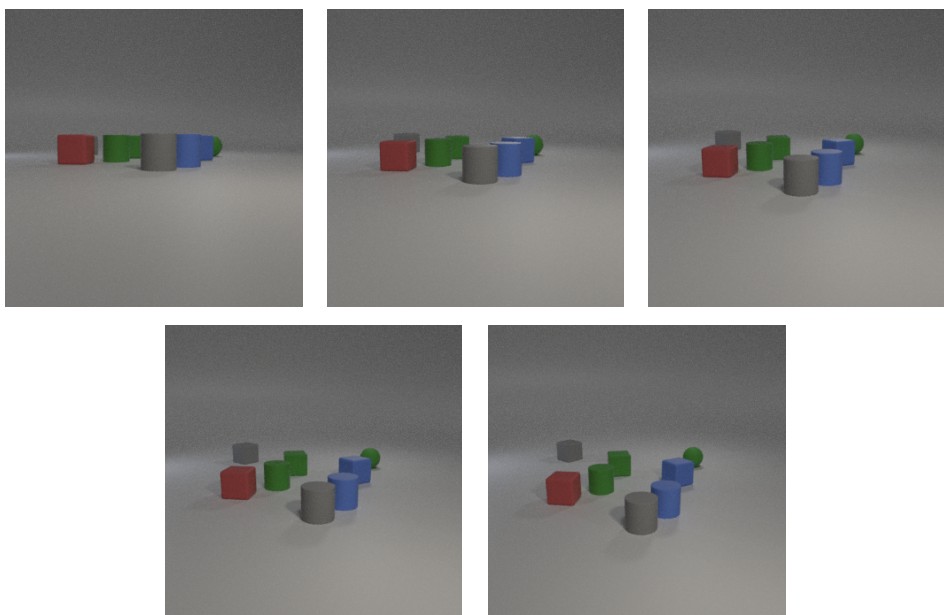

*Figure 5.* **The different occlusion levels of the CLEVR dataset, controlled by camera elevation and pitch angle.** In left-to-right, top-to-bottom order, the camera elevation and pitch angles are $(0.6, 6), (1.2, 12), (1.8, 18), (2.5, 25), (3.2, 32)$. As can be seen, in the most occluded scenario, the gray cube is almost entirely covered by the red cube, and the positioning of the green cylinder and green cube (as well as the blue cylinder and blue cube) can potentially create difficulty for binding due to their ambiguous boundaries. These issues are progressively resolved when there is less occlusion.

**Visual Genome**   Visual Genome is a large-scale vision dataset with dense annotations of objects, attributes, and relationships. From it, we construct two subsets: VG:Color and VG:TopAttr. In VG:Color, we restrict attributes to color terms; in VG:TopAttr, we retain the most frequent (top) attributes. In both cases, we keep only images containing the selected object classes and attributes, resulting in approximately 50,000 samples. We select the 20 most frequent object classes and pair them with either 20 colors or 20 top attributes. Object classes and attributes are collapsed using WordNet synsets. Attributes that are not annotated for an object are treated as an explicit feature value, in addition to the annotated attributes.

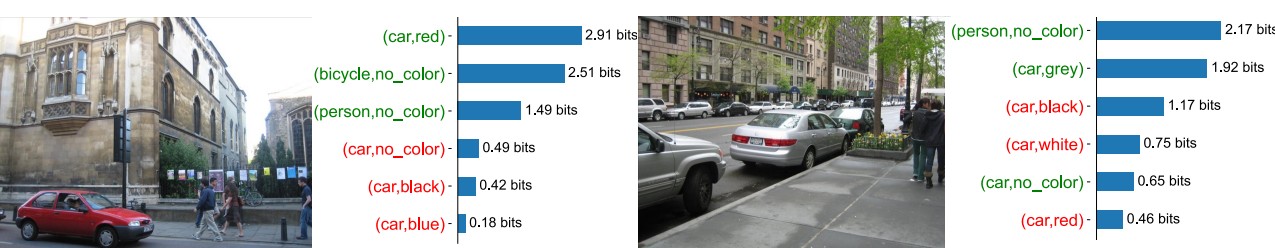

*Figure 6.* **VG:Color examples of object-code uncertainty revealed by binding probes.**

