# OpenReview forum: "Formalizing the Binding Problem"
_ICML.cc/2026/Conference — ICML 2026 regular_

### Official Review · Reviewer_KuXe · 2026-03-02

**Soundness:** 3
**Presentation:** 2
**Significance:** 4
**Originality:** 4
**Overall Recommendation:** 4
**Confidence:** 3

**Summary:**

The paper aim to theoretical derive and measure the binding problem, which is to assign a property or properties to an object or multiple ones in the scene, i.e. the attribute "red" to the object "circle". As this is a super easy task for humans with close to zero mistakes, the most enhanced VLMs are having trouble when there are more and more objects in the scene.

**Compliance With Llm Reviewing Policy:**

Affirmed.

**Key Questions For Authors:**

Questions:

* Isn't the entropy is highly affected by the number of attributes and number of objects? you can think of normalized entropy, which divide the entropy by logN, normalize by the number of entries (should be more robust to differences in the number of entries). I dont think it is the same a sdefinition 2.6.

* what can we infer from the absolute value of the result? is it have any meaning beside being relative for two cases?


While I like the paper, and the direction it goes, I feel that it is very hard to follow the mathematical parts, which are the most important part of the paper. I definitely think that a careful read of the mathematical part, with careful revision (pay attention to the questions and minor weaknesses I raised) the paper is worth publishing. You might think of moving the proofs to the Supp, and give more examples, or intuitions for the mathematical parts to make them more digestable.

**Limitations:**

As far as I understand it now, the formulation assumes we have some kind of ground truth (by having F), which in practice you may not have. Moreover, the extension for a more smooth binding, which is not one-hot is mentioned in the paper, but it seems much less trivial.

**Strengths And Weaknesses:**

Strengths:

* The paper authors tackle the binding problem which is an important task for modern vision encoders; so the issue here is timely and relevant for many tasks such as concept detection, segmentation and even for text2image.

* The paper aims to study the issue from the low-level basics, and not straight to the implementations where many times have no intuition behind them. The fact that the researchers start with first measuring the phenomenon is appreciated. Moreover, I am not familiar with a method that define the binding problem per se.

* The motivation and the introduction of the paper is well written, fluently and easy to follow. The problems arises from the binding issue are clear.

Weaknesses:

* The mathematical formulations are fairly less structured and clear than the non-technical parts: large amount of different notations, abuse of notations (i.e. on T, being used differently with superscript and subscript),  the notation of 1{wherever exists} is not familiar to me. See more on the Minor Weaknesses section.

Minor Weaknesses:
* Fig. 1 is super non-intuitive, I didnt understand it. What the background colors means? why some have gradual changing colors? The figure introduced too early, so I cant understand the notations and the formulations. It is better to start with something intuitive such as Fig. 2, and introduce Fig. 1 only when the notions are set.

* Why is the first definition is 2.1? if it is because it is under section 2.1 then you have also def 2.2 in the same section.

* Why do you term the vector in definition 2.1 a "set"? it may contain multiple instances of the same element, I guess you simply meant to a vector.

* Entropy on a vector is the common operation. What entropy on a matrix means? column/row stack it and apply entropy? is there any meaning for the shape of the matrix?

* As far as I understood, B is a matrix for the entire scene on multiple objects and F is on a single object (if so, then why without an indexing?). If so, how can we define B|F? they are applied on multiple or single object. (I guess I am missing something here). Starting from Remark 2.7 you start treat F with subindex, so it is truly confusing.

* It is better IMO to present table 2 as correlation graph. It will demonstrate more the correlation.

---

> ### Author Rebuttal · Authors · 2026-03-30
>
> Thank you for these helpful comments. We have substantially revised the notation, numbering, and exposition in the theory section (see the end for our new version). In particular, we now define $F$ explicitly as a scene-level feature-presence vector and $B$ as a scene-level binding tensor, and we added two new figures before the original figure 1 for clarity:
>
> new Figure 1: https://ibb.co/kV5Zq8JZ
>
> new Figure 2: https://ibb.co/rRF7kQXW
>
> Figure 3 (the original Figure 1 with additional captions explaining colorings): https://ibb.co/3YRBVtk4
>
> We also clarified the notation throughout. Definitions now begin at 1.xx, $F$ is referred to as a random vector rather than a set, and we explicitly explain the indicator notation $1\\{\cdot\\}$.
>
> To address the main conceptual point: both $F$ and $B$ are defined at the level of the entire scene. The variable $F$ records which feature values are present anywhere in the scene, while $B$ records which full combinations of feature values occur as objects. Thus $F$ captures marginal feature presence, whereas $B$ captures object-level conjunctions. Accordingly, $H(B\mid F)$ measures the remaining uncertainty in how present feature values are grouped into objects.
>
> We also clarified that entropy here is the ordinary Shannon entropy of the random variable $B$, regardless of whether its realizations are written as a vector, matrix, or tensor. Namely, $H(B) = -\sum_{b \in \mathcal B} P(B=b)\log P(B=b).$ The tensor shape only reflects the semantic structure of the binding variables across feature types and values.
>
> Regarding scale, we agree that $H(B\mid F)$ depends on the number of feature types, values, and objects. We view this as intended rather than problematic, since quantity is inherent to the combinatorial difficulty of the binding problem. In the degenerate case of a single object, there is essentially no binding ambiguity once the features are known.
>
> The absolute value of binding information is meaningful *in bits*, while the binding ratio measures the fraction of dataset binding uncertainty captured by the learned representation.
>
> We also clarified that we do not assume access to the ground truth of $F$, but we do define binding as the information that the representation $Z$ has on binding matrix $B$ *beyond* feature vector $F$. In fact, this is precisely true: $\mathcal{B}(Z)=I(B;Z)-I(F;Z).$ We have added this as an alternative, mathematically equivalent definition of binding.
>
> Regarding an extension to smooth features in binding, we agree that extending the framework beyond binary binding is nontrivial. We have revised the text to present it as a direction for future work.
>
> We also replaced Table 2 with a correlation graph, as suggested.
>
> ---
>
> **The following is our revision for the first few definitions of our paper, for reference:**
>
> We first specify a finite collection of feature types
> $$
> \mathcal T = \\{1,\dots,m\\}.
> $$
> For each type $t \in \mathcal T$, let $\mathcal V_t$ be the finite set of possible values of that type.
> For example, if $t$ indexes color, then $\mathcal V_t$ may be $\\{\text{red},\text{blue},\dots\\}$.
>
> $\textbf{Definition 1.1}$: Feature-presence variables and full feature code
>
> For each feature type $t \in \mathcal T$ and value $v \in \mathcal V_t$, define the binary random variable
> $$
> F_{t,v} := \mathbf 1\\{\text{some object in the scene has value } v \text{ for type } t\\}.
> $$
> The $\textit{full feature code}$ is the random vector
> $$
> F := (F_{t,v})_{t\in\mathcal T,\; v\in\mathcal V_t}.
> $$
> Thus $F$ records which feature values are present anywhere in the scene, but not which values belong to the same object.
>
>
> $\textbf{Definition 1.2}$: Binding tensor
>
> Fix an ordering of the feature types $\mathcal T=\\{1,\dots,m\\}$.
> For each tuple of values $(v_1,\dots,v_m)\in \mathcal V_1\times\cdots\times\mathcal V_m$, define
> $$
> B_{v_1,\dots,v_m}
> :=
> \mathbf 1\\{\text{some object in the scene has type-}1\text{ value }v_1,\dots,\text{type-}m\text{ value }v_m\\}.
> $$
> The $\textit{binding tensor}$ is
> $$
> B := (B_{v_1,\dots,v_m})_{(v_1,\dots,v_m)\in \mathcal V_1\times\cdots\times\mathcal V_m}.
> $$
> Thus $B$ records which full combinations of feature values occur as objects in the scene.
>
> $\textbf{Remark 1.3}$: Interpretation of $H(B\mid F)$
>
> Feature code $F$ generally does not determine the binding tensor $B$. $F$ can, however, constrain $B$. Once $F$ specifies which feature values are present in the scene, any binding involving an absent value is impossible. Thus knowing $F$ can reduce the set of feasible binding tensors. The conditional entropy $H(B\mid F)$ measures the remaining uncertainty about how the feature values are combined into objects, given knowledge of which feature values present in the scene. See the new Figure 2 for an example.

---

> > ### Author Rebuttal · Reviewer_KuXe · 2026-04-01
> >
> > I initially liked this paper, so upon precise rewriting as suggested in the rebuttal, I believe the clarity of the paper should be higher, so these clarifications and editions are resolving my concerns.

---

> > > ### Author Response · Authors · 2026-04-05
> > >
> > > Thank you for your support for our work! We appreciate your time and effort reviewing our paper.

---

### Official Review · Reviewer_rscL · 2026-03-11

**Soundness:** 3
**Presentation:** 3
**Significance:** 3
**Originality:** 3
**Overall Recommendation:** 3
**Confidence:** 2

**Summary:**

This paper proposes an information-theoretic formulation of the binding problem and develops a probing-based framework for estimating binding information in model representations. The contribution is mainly conceptual and methodological, rather than centered on SOTA benchmark performance. The paper studies binding behavior across different settings, including increasing feature complexity, occlusion, natural scenes, and out-of-distribution combinations.

**Compliance With Llm Reviewing Policy:**

Affirmed.

**Final Justification:**

Thank the authors for their efforts during the rebuttal period. I have maintained my initial rating.

**Key Questions For Authors:**

1. How directly do the authors expect the proposed binding measure to transfer to modern vision-language models?
2. How sensitive are the conclusions to the choice of probe family and feature ontology?
3. What important forms of binding failure are outside the scope of the current formulation?
4. Do the authors view this framework mainly as an analysis tool, or also as a guide for model design?

**Limitations:**

Yes.

**Strengths And Weaknesses:**

## Strengths
1. The paper is original in framing. It tries to formalize what binding means, rather than only documenting failure cases on downstream tasks.
2. The connection between the theoretical object and a practical probing framework is useful and makes the work more than a purely abstract note.
3. The empirical section is not trivial and covers several settings rather than relying on a single toy example.

## Weaknesses
1. The paper’s ambition is broader than its empirical coverage. The motivation discusses modern vision and vision-language models, but the experiments remain largely centered on Vision Transformers.
2. The conclusions still depend on the specific probing setup and feature ontology, which may narrow the notion of binding actually being measured.
3. While the framework is interesting, the paper is less convincing on how directly it will influence practical model design.

---

> ### Author Rebuttal · Authors · 2026-03-31
>
> Thank you for your thoughtful feedback. We have revised the framing and run additional experiments to address these concerns.
>
> *VLM framing*. We agree that the current empirical evidence is centered on ViTs rather than full vision-language models, and we have revised the paper to reflect that more clearly.
>
> *Sensitivity to probe family*. We agree that conclusions depend on the probe family, and we now make this dependence explicit. To study it more carefully, we added a simple routing-based probe that replaces uniform averaging of spatial tokens with a head-specific weighted mean over tokens, followed by a standard readout. Concretely, each probe head assigns different weights to spatial tokens before pooling them, allowing it to focus on the patches most relevant to the target prediction. On ColorShape (7 colors, 7 shapes), this raises the binding ratio to 95%, while differences between linear and quadratic readouts become much smaller once routing is available (Table 1). This suggests that the key missing ingredient in our earlier setup was not only decoder nonlinearity, but access to the relevant spatial tokens.
>
> Table 1: https://ibb.co/G3940FdQ
>
> We wanted to point out that this is in fact a preferable probe family not only due to its high performance, but also its similarity to the attention mechanism of transformers. The probe chooses to focus on the spatial tokens that matter the most according to the “query” (the $B_i$ or $F_i$ we are probing for). However, our probe does not require key and value projections. Instead, we only project a learned per-head “query” vector with a shared “query map”, before taking dot product directly with the spatial tokens and taking softmax for weights. Our ablations on this probe can thus provide practical insights on the various components of attention in their ability to solve binding.
>
> Ablations on this probe. Removing the shared query projection drops the binding ratio from 95% to 57%, while adding extra learned "key", "value" projections on the spatial tokens increases it further to 99% and 100%, respectively. We hypothesize that these results are due to an increasing ability to route to the precise patches relevant to the prediction, with the help of query, key, and value projections. We are examining the patch weights learned by different ablations of the weighted-mean probe to compare their routing behavior for more insights.
>
> *Sensitivity to feature ontology*. We agree that the framework is ontology-dependent by design: it measures binding relative to a chosen set of human-defined features. We view this as appropriate when the goal is to study task-relevant attributes, such as relevant object attributes in visual search tasks.
>
> *Transfer to VLMs*. Given that our new probes are highly similar to the attention mechanism, we expect our probes to naturally transfer to Vision-Language Models. Particularly, in addition to the vision-only and language-only encoders, VLMs typically contain cross-attention between vision and language. Our attention-like probe can bring insights to the various components of this cross-attention: note that the "query" of our probes (which feature combination we probe for) is analogous to the role of the language prompt embedding of VLMs.
>
> *Binding outside the current scope*. Our present formulation focuses on discrete object-feature conjunctions at a single scene level. It does not yet capture more general notions of binding such as continuous-valued features, hierarchical part-whole binding, relations between objects, or temporal binding across time.
>
> *Analysis tool vs. design guide*. At present we view the framework primarily as an analysis tool. That said, the new routing results suggest that it may also help generate architectural hypotheses, for example about the importance of routing or attention-like mechanisms in making binding information accessible.

---

> > ### Author Rebuttal · Reviewer_rscL · 2026-04-04
> >
> > Thank you for the thoughtful rebuttal. I appreciate the additional experiments and the clearer discussion of scope, probe dependence, and intended positioning of the framework. The new routing-based probe results are interesting and help clarify that accessibility of relevant spatial tokens is an important factor in the measured binding signal.
> >
> > That said, my overall assessment does not substantially change. My main concerns were about the scope and generality of the conclusions, especially beyond the current ViT-centered setting, as well as the extent to which the results depend on the probing setup and chosen ontology. The rebuttal helps clarify these points, but in my view it does not fully resolve them within the current submission. Therefore, I will maintain my original score.

---

> > > ### Author Response · Authors · 2026-04-06
> > >
> > > Thank you again for your feedback.
> > >
> > > For whether our result depends on the probing setup, we have since **conducted comprehensive ablations on probe architectures and setups, including the original ones in our paper and the ones newly introduced in our rebuttal**. Please refer to our **reply to rebuttal comments to Reviewer q2cF**, where we performed **7 different ablations** on probe setups.
> > >
> > > We have chosen the ViT since it is **the state-of-the-art vision architecture** widely used in computer vision. Beyond its standalone use, **it also serves as the backbone vision encoder for many multimodal models**, including many VLMs, VLAs, and world models.
> > >
> > > To demonstrate that **our analysis on the ViT is relevant to VLMs** and beyond,  we present an analysis of VLM attention localization, showing strong similarity to our probe's localization mechanism and indicating it may capture a key routing and localization mechanism in VLMs. We map the attention scores learned by Qwen3-VL-8B-Instruct when prompted in a binding task: “Does the image contain [color] [shape]? Answer yes or no only.”
> > >
> > > https://docs.google.com/document/d/e/2PACX-1vTzZqRt3yCpO-qfhqRuH9qpGuJIUaUa1KM06nuv0Q6Dd2mzIcfplJW35GuOwmLzxYx2ewSVo70uw4aj/pub
> > >
> > > Note here we show that **attention localization accuracy is significantly higher for correct downstream predictions than for incorrect predictions**.
> > >
> > > Overall, the strong similarity in the localization mechanism suggests that **our probe could potentially help improve VLMs by serving as a supervision or regularization signal for attention localization**, encouraging more accurate routing and object-level localization.
> > >
> > > Moreover, for models and downstream tasks **that use global representations of the ViT (e.g., [CLS], mean spatial token) such as CLIP, I-JEPA**, etc., **our mean spatial token probes can also be used as supervision or regularization signals** while training the ViT encoder in these architectures.
> > >
> > > We hope our response addresses your concerns. We would also greatly appreciate your reconsideration of our score. And thank you, again, for taking the time and effort to review our paper; we find your feedback very valuable in improving our work.
> > >
> > > [1] https://arxiv.org/abs/2603.14482
> > >
> > > [2] https://openreview.net/forum?id=neMAx4uBlh
> > >
> > > [3] https://arxiv.org/abs/2601.09322
> > >
> > > [4] https://arxiv.org/abs/2503.08723

---

### Official Review · Reviewer_6yp4 · 2026-03-12

**Soundness:** 4
**Presentation:** 3
**Significance:** 3
**Originality:** 3
**Overall Recommendation:** 5
**Confidence:** 3

**Summary:**

This paper investigates the "binding problem" in visual recognition  , which addresses how a system correctly attributes distinct features to the same object. Given that current vision models (such as ViTs  ) frequently misattribute features in multi-object scenes, this work proposes an information-theoretic framework to formalize the problem, defining binding as the reduction of uncertainty in object combinations given the available features. The study introduces a probing method to quantify binding information within model representations and evaluates pre-trained models across synthetic and natural datasets featuring varying feature counts, occlusions, and out-of-distribution combinations.

**Compliance With Llm Reviewing Policy:**

Affirmed.

**Final Justification:**

My main concerns were clarified, so I retained my positive rating.

**Key Questions For Authors:**

There is no obvious problem. This is a paper with certain research significance.

**Limitations:**

yes

**Strengths And Weaknesses:**

1. Clear problem definition and motivation: defining the binding problem, which has long existed in cognitive science and visual research, with a clear information-theoretic measure holds significant theoretical importance;
2. The proposed methodology is logically rigorous and well-justified: to quantify binding information, the authors introduce a highly practical estimator. This mechanism hinges on conditional entropy decomposition and leverages a probe model to approximate the conditional distribution, ultimately yielding a robust estimation of the binding information;
3. The writing is clear and the experiment is well-defined.

---

> ### Author Rebuttal · Authors · 2026-03-31
>
> Thank you for your positive feedback! Please let us know if there are any additional points of clarification required from us.

---

> > ### Author Rebuttal · Reviewer_6yp4 · 2026-04-03
> >
> > I think this paper is really interesting and look forward to the author 's further research. Thus, I maintain my score.

---

> > > ### Author Response · Authors · 2026-04-05
> > >
> > > Thank you for your support for our work! We appreciate your time and effort reviewing our paper.

---

### Official Review · Reviewer_q2cF · 2026-03-12

**Soundness:** 3
**Presentation:** 2
**Significance:** 3
**Originality:** 3
**Overall Recommendation:** 4
**Confidence:** 4

**Summary:**

The paper investigates the feature binding problem in vision transformers, mostly in the context of CLIP and DINO models. It builds on observations of object-identity binding in patches and focuses on feature binding in CLS and patch tokens. To study this, the paper starts by formalizing the problem: it divides it into feature- and binding-codes / tensors. These are the main objects of study, with the former expressing the features present in the scene (such as colors and shapes), and the latter expressing the objects themselves, e.g. how particular colors and shapes are associated with each other. The main proposal is to understand binding ability through the binding information metric, defined as the reduction in the uncertainty of bindings given the features once the representation is also provided. Since estimating this quantity directly is computationally intractable, the paper provides an approximation by estimating the necessary conditional distributions through approximations involing quadratic probes, teacher forcing, and weight sharing. Empirically, the proposed score is then used mainly for interpretability, with most of the models (DINO, CLIP) showing broadly similar behavior of scores around 0.5

**Compliance With Llm Reviewing Policy:**

Affirmed.

**Final Justification:**

The rebuttal addressed my main concerns regarding the motivation and technical contributions, and the work is significantly stronger as a result.

**Key Questions For Authors:**

See above

**Limitations:**

societal: yes
general: partly; I think discussing train/test, model selection, and the dependence on the chosen probe family would be useful.

**Strengths And Weaknesses:**

Strengths:
1. The problem of binding is interesting
2. The approach of the problem is great conceptually: I think that decoupling feature and binding predictions and expressing the binding information as the reduction of uncertainty is very useful. This is also appreciated because binding is often discussd in the context of prediction accuracy, but taking into account the distribution of prediction is potentially more useful.

Weakenesses:

1. It is unclear whether the probe training uses proper train/test splits, or if the reported estimates are affected by overfitting. As far as I understand, if the current quadratic probes were replaced, say, by a DNN, I'd expect all of the results to suddenly look better - in other words, a DNN with sufficient capacity could overfit to the whole dataset and result in $H(B | F) = H(B | F, Z)$, yielding a perfect binding score. Having a validation set would address this issue; it would also (partly) address the following point.
2. Relatedly to the first point, I do not find the motivation for non-linear probing sufficiently established. The paper argues that linear probes are insufficient because binding is combinatorial, but then introduces an elaborate estimation pipeline involving teacher forcing, quadratic probes, and weight sharing. This makes it near impossible to tell which parts contribute to the binding scores. Because the proposed pipeline is complex, the justification for each part of it should be clear and demonstrated. Currently this is not the case.
2. I think the treatment of linear probing is too dismissive. The paper states that prior works rely primarily on linear probes and argues from this that linear probes are insufficient. I understand the intuition, but I do not think linear probing should be hand-waved away so quickly. For example, Koishigarina et al. (which the authors cite) argue that binding information can be recovered through linear probing. So why not consider this as an option?
3. While I appreciate the information-theoretic approach to the problem, I keep wondering why simpler prediction-based approaches couldn't have been used or compared against. In order to assess the binding ability in vision models, one would compare feature-probing and binding-probing accuracies. This work argues that we need to take into account more information than just the accuracies -- and that makes sense. So wouldn't the next natural step be comparing other statistics of the distribution, say, entropy? Instead, this work skips these steps and jumps to uncertainty reduction due to the presence of a representation. I think a slower progression from current practice to the proposal would be appreciated.
4. Several important ablations seem missing: a comparison of different probe types (with an extreme being a DNN with a few layers), a rank ablation for the quadratic probes, validation-based probe selection analysis (and how that influences the results), and an ablation using only the CLS token rather than concatenating CLS with averaged spatial tokens -- though the last one is optional.


Minor:
1. I don't understand what purpose Theorem 2.12 serves -- it's not about the estimation of the binding information, its introduction also seems abrupt. I'd appreciate additional context and intuition.
2. L271–274: If both matrices are of rank $r$, I don't see how the computations differ.
3. I coulnd't follow this sentence: "Namely, for each feature value we learn a single set of weights, and reuse these projections across all bindings containing that feature, while parameters involving the condition B<i are not shared" - what kind of projections are considered here?
4. Relatedly, L263: How is $x \in \mathbb{R}^d$?  don't larger steps result in a larger input?

---

> ### Author Rebuttal · Authors · 2026-03-31
>
> Thank you for these really helpful feedback. We have added new experiments and clarifications addressing probe generalization, probe family, and the role of linear versus nonlinear decoding.
>
> *Generalization / overfitting*. The reported probe results are already evaluated on held-out data with disjoint $B$ values; we will make this explicit. In addition, we created a stricter test split with disjoint $F$ values as well, and observed little-to-no train/test gap for both the original probes and the new probe family below. This indicates that the probes are indeed learning (generalizable) binding structures in the representations.
>
> *Probe family and ablations*. We added a new weighted-mean probe. Instead of averaging all spatial tokens before probing, it computes a simple head-specific weighted average over spatial tokens, concatenated with CLS, and then applies a linear, quadratic, or DNN readout. It performs a simple routing mechanism (by weighting patches differently) and avoids the information loss of uniform averaging. It also bears similarity with “attention” where the probe chooses to focus on the spatial tokens that matter the most according to the “query” (the $B_i$ or $F_i$ we are probing for). However, our probe does not require key and value projections. Instead, we only project a learned per-head “query” vector with a shared “query map”, before taking dot product directly with the spatial tokens and taking softmax for weights. (See below for ablations on this probe.)
>
> Empirically, this probe substantially improves binding decoding on ColorShape (7colors7shapes) (see Table 1), reaching 95% binding ratio, whereas the choice among linear vs quadratic readout matters much less once routing is available. This suggests that the main missing ingredient in the previous setup was not higher-order decoding alone, but the ability to route to relevant spatial tokens.
>
> Table 1: https://ibb.co/G3940FdQ
>
> *Why not only linear probing?* We agree that linear probes should not be dismissed, and we will revise the text accordingly. In fact, with the weighted-mean representation, linear readout already performs very strongly (Table 1). Our conclusion is therefore more precise than before: linear readout can be sufficient once the relevant information is properly routed. Note that without routing, linear probes perform decidedly worse than quadratic probes.
>
> *Why not compare to simpler prediction-based approaches?*
> We agree that a slower bridge from current practice is useful. We will make this clearer by explicitly relating our measure to prediction-based quantities: we will show that a mathematically equivalent definition of binding is in fact $\mathcal{B(Z)}=I(B;Z)-I(F;Z)$, meaning that our definition is precisely the information about bindings beyond information about features in the representation. Empirically, we now also report the underlying $F$ and $B$ accuracies alongside the binding ratio, so the connection to standard prediction-based evaluation is transparent.
>
> *Missing ablations.*
> We have now added:
>
> (1) probe-family comparisons (linear, quadratic, DNN) on the readout (Table 1),
>
> (2) validation/test split analysis,
>
> (3) Ablations on the weighted-mean probe: removal of the "query" projection shared across heads and using only a learned "query" vector per-head. The binding ratio drops from 95% to 57%. We do find, however, that *adding* "key" and "value" projections for the spatial patches improve performance to 99% and 100% respectively on the binding ratio. We hypothesize that these results are due to an increasing ability to route to the precise patches relevant to the prediction, with the help of query, key, and value projections respectively. To verify this hypothesis, we are examining the "attention scores" (patch weights) learned by different ablations of the weighted-mean probe to compare their routing behavior for more insights.
>
> We also tested teacher forcing ablations on the weighted-mean probe and found little-to-no difference in performance.
>
> *Theorem 2.12*. This is an application of the data processing inequality in information theory on our definition of binding information. It demonstrates that internal processing of the input image cannot increase binding information (if some information is lost in one layer, then it cannot be recovered in subsequent layers.) A sidenote is that decodable binding information can increase across layers, because internal processing can make binding more decodable.
>
> *Minor points.*
> The statements around L263 and L271–274 were indeed typos, and have been corrected. We have also rewrited the explanation of the old weight-sharing scheme more concretely; however, this is less central now because the simpler weighted-mean probe has a much higher performance than the quadratic probes, indicating the importance of weighting/routing in decoding binding.
>
> Since the weighted-mean probe is new, we will be running this family of probe across different datasets as before.

---

> > ### Author Rebuttal · Reviewer_q2cF · 2026-04-03
> >
> > Thank you for the detailed response. I find the conceptual contribution interesting, but the rebuttal introduces a change that undermines the paper's own claims and prevents me from resolving my issues.
> >
> > **The rebuttal changes what $Z$ is in the formalism, and therefore what is being measured.** The formalism defines $\mathcal{B}(Z) = I(B; Z | F)$ for a given representation $Z$. In the original paper, $Z$ is CLS + mean-pooled spatial tokens ( essentially a global vector) and the paper explicitly argues that binding "should be measured from a global representation of the input" (Section 3.1). The ~50% binding scores are a meaningful statement about what the model encodes globally. However, the rebuttal effectively redefines $Z$ as the full set of spatial tokens accessed through learned query-key-value routing over patches. This answers a different question: not "does the model's representation encode binding?" but "can binding be recovered from distributed patch features by a sufficiently expressive decoder?"
> >
> > Indeed, similar probes have been established in Kang et al. (2025, "Is CLIP ideal? No. Can we fix it? Yes!"), who showed that compositional information can be recovered from CLIP's patch tokens with a learned cross-attention decoder, even though CLIP's pooled representation fails at compositional tasks. The rebuttal probe is functionally the same approach, presented as a measurement (though it redefines what $Z$ is). That is fine in principle, but it becomes a question about the total binding information in the spatial embeddings, not about "global representations" whose failure cases actually motivate the study.
> >
> > The above is a major point and I am not sure how to resolve it in a satisfying way. The following is another insufficiently addressed point:
> >
> > - The claim that probes generalize to held-out data with disjoint feature combinations would be meaningful if supported, but the rebuttal provides no numbers. A verbal assurance is insufficient for a claim the paper's validity depends on. Though I appreciate that a validation set was used.
> >
> > I would encourage strengthening that analysis with proper generalization evidence and baselines, rather than introducing a probe that changes the question being asked (or changing the motivation for the paper? I'm not sure which one is more valuable). I maintain my score for now.

---

> > > ### Author Response · Authors · 2026-04-06
> > >
> > > Thank you again for your feedback. We agree that this is a very valid distinction, in terms of what the $Z$ being measured is. We wanted to argue that it is valuable to measure binding information **under both scenarios**, i.e., when $Z$ is a global representation, and when $Z$ includes all spatial tokens.
> > > * Models like CLIP typically use a global representation such as the [CLS] token, especially for tasks such as classification and recognition. This is indeed the original motivation of our paper using the global rep.
> > > * On the other hand, models like the VLM use *all spatial tokens* in their processing to arrive at their downstream prediction, so if we want to motivate a study of these architectures, then we should use an attention probe such as ours that take in all spatial tokens.
> > >
> > > We thus **conduct comprehensive ablations on both kinds of probe architectures, including the original ones in our paper and the ones newly introduced in our rebuttal**. We perform the full set of ablations as adviced in your original response, on train/val/test sets.
> > > - The val set contains held-out $B$ vectors: no binding codes in the val set appear in training.
> > > - The test set contains held-out $F$ vectors (and thus also $B$ vectors): no feature (or binding) codes in the test set appear in training or validation.
> > >
> > > **Global representation probes (CLS + mean spatial) ablations (5 ablations):**
> > >
> > > https://docs.google.com/document/d/e/2PACX-1vQwtVoeZOCYCUOPwQme8U2xW20eoD96q5KxqXDkBEbMtRhd3jpbkcKBdYwf-5wJegxqXYrDlyG5eK4g/pub
> > >
> > > We note that the ablations above are conducted on the original setup in our paper (grayed). That is, linear feature probe and quadratic binding probe with teacher forcing, weight sharing, and rank = 32. Feature probes refer to that for $H(F|Z)$, while binding probes refer to that for $H(B|Z)$. Binding info/ratios are omitted because they are dependent upon the choice of the feature and binding probes, and are derived from their results directly.
> > >
> > > The ablations here show little sign of overfitting, both in terms of validation and testing. Note that using a linear probe for binding results in significant decrease in performance (~8%), suggesting that quadratic probes are indeed preferred for binding. Reducing the quadratic probe rank to r=8 slightly hurts performance (about 3%), while expanding to r=128 has minimal effect. For the DNN probes with significantly more parameters than the baseline probes, there is minimal improvement, indicating that linear feature and quadratic binding probes may better reflect the structure of binding information than DNN probes. There is slight improvement in performance when training the full binding probe without the weight sharing scheme (about 2%), but given the massively reduced number of parameters (and therefore higher efficiency), we consider weight sharing a valid option for probing. When ablating teacher forcing, we find that the performance decreases by about 5%, indicating that teacher forcing (conditioning on the ground truth bindings) is relatively important, also considering the fact that it is part of the binding formula (eq. 4 in our paper). We also observe negligible decay in performance (about 1%) when using CLS only for our probes, suggesting that mean spatial tokens may yield little additional benefit to decoding binding, which suggest that using only the CLS as the global representation may be a valid option (which reduces parameters as well). **Overall, these ablations show that our original probing setup is robust across key variants.**
> > >
> > > **All token weighted-mean probes ablations:**
> > >
> > > https://docs.google.com/document/d/e/2PACX-1vQN7ADIBQZQoxnG6HrrgVIvvjV9cnutuLR5gv-iakWJ_7FUKDv8ZOv79SmGmd8KKJneFlS74psmuFNw/pub
> > >
> > > We note that prior work has explored using attention-like probes on language and vision transformers, but theirs are typically full, multi-block transformer probes [1, 2, 3, 4]. In contrast we train a simplified query-only probe to understand binding in terms of routing, especially for models using all spatial tokens.
> > >
> > > To supplement our rebuttal, **we analyze attention maps from the weighted-mean probes** and **find that they almost always localize the target object when it is present**. When it is absent, they often attend to objects sharing its color or shape, and sometimes to other patches. Qualitative and quantitative results are shown in the figures here: https://docs.google.com/document/d/e/2PACX-1vTDaw-dx5IVtVcUDrTxEjK3im39z6aVGFfDJKDI5O67saeiDWSGGpMKU8hxwB2IuXZt6m1aVO6F1nHp/pub
> > >
> > > We hope our response addresses your concerns. We would also greatly appreciate your reconsideration of our score. And thank you, again, for taking the time and effort to review our paper; we find your feedback very valuable in improving our work.
> > >
> > > [1] https://arxiv.org/abs/2603.14482
> > >
> > > [2] https://openreview.net/forum?id=neMAx4uBlh
> > >
> > > [3] https://arxiv.org/abs/2601.09322
> > >
> > > [4] https://arxiv.org/abs/2503.08723

---

### Decision · Program_Chairs · 2026-04-30

**Decision:**

Accept (regular)

**Comment:**

The reviewers and I agree that this paper contributes an original and useful conceptual framing of the binding problem.
Concerns about clarity of the theory and the details of the probing methods were raised during the review and addressed by the authors.
After skimming the paper and carefully reading the discussion, my impression is that the concerns about the soundness of the probing have been sufficiently addressed.
Therefore I consider this paper a solid contribution and recommend accepting it to the conference.